# Willingness to Pay for Condoms among Men in Sub-Saharan Africa

**DOI:** 10.3390/ijerph16010034

**Published:** 2018-12-24

**Authors:** William Evans, Kuyosh Kadirov, Ibou Thior, Ramakrishnan Ganesan, Alec Ulasevich, Bidia Deperthes

**Affiliations:** 1Milken Institute of Public Health, The George Washington University, Washington, DC 20052, USA; 2United States Agency for International Development, Washington, DC 20523, USA; kkadirov@usaid.gov; 3John Snow Incorporated, Arlington, VA 22209, USA; Ibou_thior@jsi.com; 4Abt Associates, Rockville, MD 20852, USA; ram_ganesan@abtassoc.com; 5Ulasevich Social Science Research, Silver Spring, MD 20902, USA; alec.ulasevich@gmail.com; 6United Nations Family Planning Agency, New York, NY 10158, USA; deperthes@unfpa.org

**Keywords:** HIV/STIs, condoms, willingness to pay, sub-Saharan Africa, social marketing, branding

## Abstract

HIV/AIDS and other sexually transmitted infections (STIs) continue to be among the greatest public health threats worldwide, especially in sub-Saharan Africa (SSA). Condom use remains an essential intervention to eradicate AIDS, and condom use is now higher than ever. However, free and subsidized condom funding is declining. Research on how to create healthy markets based on willingness to pay for condoms is critically important. This research has three primary aims: (1) willingness of free condom users in five African countries to pay for socially marketed condoms; (2) the relationship between specific population variables and condom brand marketing efforts and willingness to pay; and (3) potential opportunities to improve condom uptake. Nationally representative samples of at least 1200 respondents were collected in Kenya, Nigeria, South Africa, Zambia, and Zimbabwe. We collected data on a range of demographic factors, including condom use, sexual behavior, awareness of condom brands, and willingness to pay. We estimated multivariate linear regression models and found that free condom users are overwhelmingly willing to pay for condoms overall (over 90% in Nigeria) with variability by country. Free users were consistently less willing to pay for condoms if they had a positive identification with their free brand in Kenya and Zimbabwe, suggesting that condom branding is a critical strategy. Ability to pay was negatively correlated with willingness, but users who could not obtain free condoms were willing to pay for them in Kenya and Zimbabwe. In a landscape of declining donor funding, this research suggests opportunities to use scarce funds for important efforts such as campaigns to increase demand, branding of condoms, and coordination with commercial condom manufacturers to build a healthy total market approach for the product. Free condoms remain an important HIV/AIDS prevention tool. Building a robust market for paid condoms in SSA is a public health priority.

## 1. Introduction

HIV/AIDS and other sexually transmitted infections (STIs) continue to be among the greatest public health threats worldwide. Sub-Saharan Africa (SSA) faces a disproportionate burden, with some 70% of the global infections [1,2]. Young people, especially girls and young women age 15–24, are at the greatest risk [3]. Given the dramatic increase in youth and young adult populations in SSA (some 250 million will be under age 25 by 2030), and their potential sexual risk taking behavior, the HIV/AIDS epidemic remains a public health priority calling for comprehensive and innovative solutions [4].

Condom use has been and remains an essential intervention in the ongoing struggle to eradicate AIDS. In addition to being highly effective in preventing the spread of HIV, condoms are also relatively inexpensive at just US$0.03 per unit [5], and cost-effective in their dual protection against STIs and unintended pregnancies. As a result, donor programs and the international AIDS community have called for, and encouraged, promotion of condom use in countries with high HIV prevalence.

As a result, condom use is now higher than ever before [6,7,8], and has prevented 50 million HIV infections since the 1980s [9]. Despite this progress, key gaps remain in some countries, where condom use has stagnated or decreased. Condoms in many countries in sub-Saharan Africa are provided for free by the public sector, which is often heavily dependent on donor support. In addition to being unsustainable in the long term, a condom supply that is heavily driven by donors can impede the private sector from playing an active role in the condom market. In many countries for instance, young people find commercial condoms more appealing, easier to access, or of better quality than free condoms offered in the public sector. A total market approach that encompasses public, nonprofit, and for-profit actors is considered critical to leveraging the unique capabilities of private sector marketers, social marketing organizations (SMOs), and the public sector.

Social marketing (i.e., the use of marketing theory, skills and practices to achieve social and behavioral change) is a widely used strategy in sub-Saharan Africa (SSA), and worldwide, to create and sustain markets for condoms based on subsidized pricing [10]. Social marketing nongovernmental organizations (SMO) have successfully branded, promoted, and distributed large numbers of condoms in SSA and thereby increased demand for the product [11]. Social marketing of condoms is widely viewed as an important strategy to combat the spread of HIV/AIDS [12].

Social marketing has been used to create positive identities for condom brands, increasing demand [13]. By creating a positive identify for a condom brand, consumers develop a relationship with the product and recognize the lifestyle benefits it confers [14]. Previous research suggests that such positive identification with condom brands can promote greater willingness to pay, and increase the opportunities for donors, governments, and SMOs to effectively grow the condom market by encouraging purchasing among those able to pay [11].

Willingness to pay has been defined as the maximum price at or below which a consumer will definitely buy one unit of a product [15]. Willingness to pay for products is important to social marketers and the field of development because it helps to shape demand creation strategies [11]. While free distribution of products such as condoms is a key strategy, and is both ethically responsible and part of effective demand creation, appropriate pricing of subsidized and full-cost condoms for those who are able to pay can help to create a robust and healthy market that maximizes demand [16].

There are three widely used methods for measuring and evaluating willingness to pay: bidding game, discrete choice model, and the Van Westendorp price sensitivity measure [17,18,19,20]. In the current study, analyses were based on the bidding game data.

The overall aims of the current study are to evaluate: (1) Willingness of free condom users in five African countries to pay for socially marketed condoms; (2) the relationship between specific population variables and condom brand marketing efforts and willingness to pay; and (3) potential opportunities to improve condom uptake and needs for future programs and research.

This study was designed to answer two primary research questions (RQ): RQ1) If the supply of free condoms were reduced or restricted, would people who currently use these condoms purchase priced condoms or discontinue using condoms? RQ2) If the price of condom brand(s) were increased, would those who use these condoms discontinue using condoms, or would they switch to other condom brands? Additionally, for purposes of future research, we ask what actions should be taken to improve the health of condoms markets in SSA, including commercial brands.

## 2. Methods

### 2.1. Design

The study was a single time-point, nationally-representative, cross-sectional survey in Kenya, Nigeria, South Africa, Zambia, and Zimbabwe. The surveys were carried out in randomly selected geographic areas, where quantitative data were collected from adult men who purchased or obtained a condom in the three months preceding the surveys. The desired sample of participants per country was set at 1200, with quotas for urban versus rural respondents; and brand types that a user most often used (i.e., free, socially marketed (SM), and commercial). Ethical approval was obtained from nationally recognized ethics boards in Kenya and Nigeria in March 2017 (AMREF—ESRC P305/2017). The surveys in South Africa, Zambia, and Zimbabwe were granted exemption from local ethical approval by the national ethics boards (IRB #17-006).

The surveys collected data on all three approaches described earlier to evaluate willingness to pay for condoms. For purposes of analyses reported here, we focused on the bidding game data because it provided a straightforward way to model willingness to pay in a multivariate analysis. Specifically, we used a bidding game approach to address the question what would free condom users do if the supply/availability of free condoms was reduced/restricted. Accordingly, questions relevant to this approach were only applied to respondents who said that they used free condoms most often. For the bidding game approach, researchers asked users of free condoms a set of questions about the lowest-priced SM condom brand in the country. They were first asked of their willingness to pay the current median price for the lowest-priced SM condom in the country (in local currency). Based on their response, they were given additional questions to estimate how much a respondent would “bid” for such a condom. The questions were as follows. (1) Would you be willing to pay X for [BRAND]? (2) If Yes to #1: Would you be willing to pay X +25% for [BRAND]? (3) If No to #1: Would you be willing to pay X −25% for [BRAND]? (4) If Yes to #2: Would you be willing to pay X +50% for [BRAND]? (5) If No to #2: Would you be willing to pay X −50% for [BRAND]? (6) To all: What is the maximum amount you would be willing to pay for [BRAND]?

The analysis of these questions yielded a demand curve showing the proportion of respondents willing to buy the brand at or below different price points. The bidding game approach has advantages due to its simplicity (i.e., a set of direct questions) and because it produces reasonable estimates of price thresholds for a product—the threshold being the price above which price would significantly affect demand. However, the approach does have limitations. Some experts believe its application is likely to underestimate the lower threshold, thus yielding a low estimate of a population’s willingness to pay. It is also considered unsuitable for assessing price sensitivity of a brand in a context where consumers have multiple choices in brands and prices.

### 2.2. Sampling

The sampling approach in each country was to attain a nationally representative sample of men 18 years of age or older who used a condom in the three months preceding the survey. Quotas were used to ensure inclusion of an equal number of urban and rural respondents, and of respondents who primarily used free, SM, or commercial condoms. The sampling plan assumed that most users of commercial brands lived in urban areas, so no quota was set for rural users of commercial brands.

The study used a multistage cluster sampling approach, whereby each country was first divided into a given number of clusters, based on existing administrative boundaries within each country. Each cluster was then assigned a sample size requirement proportional to the target population (i.e., males 18 years and above) in each cluster as a percentage of the total target population for the country as a whole. This ensured appropriate distribution of the 480 rural and 720 urban respondents for each country’s desired sample. Clusters that were entirely urban (e.g., Nairobi) were not assigned a sample for rural respondents.

Within each primary sampling unit (PSU) visited, the data collection team randomly selected a starting point from a list of at least three prominent landmarks (e.g., school, police station, hospital, and house of worship). From the selected landmark, the closest household in a northeasterly direction was selected as the first household to approach for an interview. After the first household was complete, collectors walked northeast, skipping four households to then include the fifth as part of the sample. Households were contacted until the appropriate number of eligible respondents were interviewed within the PSU. Respondents were considered eligible if, at the moment of contact by the data collection team, they were 18 years of age or older; used a condom in the previous three months; qualified for a quota segment that was not yet complete; and provided verbal consent. Households within each PSU were selected randomly based on proximity to key landmarks (Police station, hospital or house of worship). The sample was weighted by demographic characteristics of the respondents (age, socioeconomic status, and urbanicity) to adjust for population parameters.

### 2.3. Measures

The following measures were included in the survey:AgeMarital statusNRS (National Review Survey) Social Grade (Kenya, Nigeria, Zambia, and Zimbabwe)Living Standard Measure (LSM) (South Africa only)Residence (urban or rural)Self-reported condom use with most recent partnerSource of condom used with most recent partner (free, socially marketed, or commercial)Distribution source of most recent condom (where obtained)Condom brands used (price established independently by market analysis)Brand attributes (measure of how favorably the respondent viewed a particular condom brand)Willingness to pay (bidding game)

The questionnaire based on these measures was programmed for computer-assisted in-person survey administration.

### 2.4. Data Collection

Data collection in the five target countries took place in April and May of 2017. Additional data collection was conducted in Kenya in July of 2017. Kantar Public conducted the field work and was responsible for recruiting, training, and overseeing the study’s enumerators. To facilitate data collection, field teams used Android-based Samsung tablets equipped with NIPO Nfield 6.0 to facilitate data capture during face-to-face interviews with participants. A pretest of the methodology and collection instrument was undertaken in all five countries before collection began.

Each data collection team comprised data collectors and a supervisor in the field who reviewed and verified data collected by their team. Additional support was provided by a field manager who spent half-time in the field and the other half in the office. An independent Kantar quality control team ensured the overall quality of survey data. All interviews were conducted with the support of Android-based Samsung Galaxy tablets, with data captured through direct entry into the NIPO Nfield application. At the close of each interview, the interviewer reviewed the completed survey to ensure that all questions had been answered. The interviewer then sought clarifications as needed before moving to another interview or leaving the area. Supervisors reviewed data captured by field teams at the end of each day, and made the final determination of the completeness and overall quality of data submitted by team members. Any corrections were ultimately reflected in the centralized database.

### 2.5. Analysis

IBM’s SPSS Statistics 25.0 (IBM, Armonk, NY, USA) was used to conduct all analyses. We calculated descriptive statistics and bivariate statistics for condom use, demographics, and willingness to pay. We then estimated two sets of linear regression models. The first set of models included all users of free condoms defined as participants who were asked DC set of questions, the designation was checked against usual brand of condoms used by that individual.

The second set of models included only free condom users who were aware of socially marketed brands (‘Trust’ brand in Kenya and ‘Protector Plus’ brand in Zimbabwe). We only had the relevant variable for brand awareness available in Kenya and Zimbabwe due to variability in the survey administration in the other countries, and thus limited the second analysis to these countries.

The dependent variable in these analyses was the highest price the individual was willing to pay for SM brand of condoms based on the discrete choice series of questions. The range of price values for this variable was based on a multiple of the actual price of the condom available in the specific country, with values ranging from 1.5× price, 1.2×, 1×, 0.8×, 0.5× and 0.

Models included several other potential predictor variables:Willingness to buy: Participant answered that if free condoms are not available they would purchase condoms (coded as 1 for yes and 0 for all other answers).Socioeconomic category: Calculated 5-category SEC with less affluent class coded with a higher numerical value (1 to 5).Time to obtain condoms: Coded usual length of time reported to buy/obtain the respondent’s usual condom brand into five categories (0 to 10 minutes, 11 to 20, 21 to 30, 31 to 40, and over 40).Free condom brand attributes: Sum of yes (1) responses to a set of 11 specific brand attributes excluding brand attributes pertaining to perception of marketing activities and price (e.g., has good price promotion and comes in affordable pack sizes). A yes response indicated a positive perception of the brand on specific brand attribute.Socially marketed condom brand attributes (used only for analyses that included participants aware of the socially marketed brand, same coding as for free brands).Rank order of free and socially marketed condom brand attributes: if the sum of attributes for socially marketed condoms was greater than for free condoms, participants received a code of 1 indicating more positive perception of the socially marketed brand (otherwise they received a score of 0 indicating preference for free condoms brands or no preference). This approach was based on the work of Bass and Tarzyk [21] and Hofmeyr, Goodall, Bongers, and Holtzman [22], who recommended ordinal rankings of brand preferences.

## 3. Results

The five countries had similar sample compositions with some exceptions. In each country the sample was mostly urban (64% in Kenya, 67% in Nigeria, 62% in Zambia, and 60% in Zimbabwe), with the highest percentage of urban sample in South Africa (82%). In most countries, the sample was age 18–34 (77% in Kenya, 76% in Nigeria, 75% in Zambia, and 73% in Zimbabwe). However, in South Africa, there were more respondents over 35 (52%). There was a higher percentage of single respondents in Nigeria (71%), Zambia (66%), and South African (65%) in comparison to Kenya (60%) and Zimbabwe (58%). Among the four countries that provided data on NRS Social Grade, the percentage of respondents in the lowest socioeconomic classes (C2, D, and E) was approximately 60% in Nigeria, Zambia, and Zimbabwe, but 71% in Kenya. South African data included the Living Standard Measure (LSM), and respondents in the lowest socioeconomic classes (LSM 6 or below) represented 64% of the sample.

We observed considerable variability in the source of condoms in each of the five countries, shown in Figure 1. The percentages were based on weighted frequencies in each country with unweighted frequencies presented in the Table 1. In Kenya, Nigeria, and Zimbabwe, socially marketed condoms were the most widely used. In South Africa (SA), freely available condoms were overwhelmingly the most common (at nearly 86% use), and in Zambia it was closely divided between free (47%) and socially marketed (46%). Figure 1 also displays rates of commercial (comm) condom use by country.

To answer RQ1, we examined self-reported actions that free condom users would take if free condoms were unavailable. Substantial majorities in Kenya (69.9%), Nigeria (69.9%), and Zambia (62.4%) indicated that they would purchase condoms at a nearby store. In Zimbabwe, respondents were almost evenly divided between purchasing nearby (38.4%) and going to another location where condoms were freely available (40.8%). Only in South Africa, which has a very large number of free condom users, did respondents indicate that they would go to another location to obtain free (63.2%).

Percentages are based on weighted data; unweighted frequencies are presented in the table.

To begin to answer to RQ2, we plotted free users’ willingness to pay for condoms using a measure of price for the paid condoms (socially marketed brand price used as the referent) available in market. We created a standard (adjusted) measure of price across the five countries to ensure comparability. Figure 2 shows that free users are (1) willing to pay at least 0.5 of the standard price for the local socially marketed condom brand (over 50% in Zimbabwe, 60% in Zambia, near or over 90% in Kenya, Nigeria, and South Africa). As price increases, as expected we see declining percentages of individuals willing to pay. However, at 1.0 of usual price, more than 90% in Nigeria, 80% in Kenya, and 70% in South Africa were willing to pay. At 1.5 times the usual price, over 75% in Kenya were still willing to pay, and nearly 50% in Nigeria and South Africa.

To further examine RQ2, we estimated two sets of linear regression models in Kenya and Zimbabwe (Table 2 and Table 3). For each country, we estimated the willingness to pay among those who most recently used free condoms (*N* = 589 in Kenya and *N* = 500 in Zimbabwe). Additionally, in order to explore the potential influence of the brand perception of the socially marketed brand, linear regressions were estimated on a subset of users of free condoms were aware of a socially marketed brand (*N* = 490 in Kenya and *N* = 403 in Zimbabwe). For all models, the dependent variable was the highest price the respondent was willing to pay and indicators for wealth, willingness to purchase condoms if free condoms were not available, travel time to obtain free condoms (proxy for convenience), and the sum of free condom brand attributes were entered as predictors. For the regressions for the subset of free condom users who reported awareness of the SM brand, the sum of SM brand attributes was entered as a predictor. A separate regression model used the rank order score of sum of brand attributes as predictors instead of sums of brands equity for free and socially marketed brands. For all models, we examined multicollinearity between predictors. Weighted data were used in the analyses.

In both countries, not surprisingly we found that those who were less affluent (and therefore less able) were less willing to pay. However, further supporting an affirmative answer to RQ2, we found users were willing to pay more if no free condoms were available. In Kenya, free condom users were willing to pay if free condoms were not conveniently available (too far away). However, we found the opposite in Zimbabwe.

Finally, in both countries we found an inverse relationship between perceived brand attributes of the in-country free condoms and willingness to pay: The more positive the free brand attributes were perceived to be, the less willing users were to pay for condoms. In other words, those who were satisfied with their free condom brand saw less incentive to switch to a paid alternative.

Among those aware of the socially marketed brand, the brand perception of the alternative brand (SM brand) was a significant predictor only in Zimbabwe. Specifically, greater preference for the SM brand over the free brand was positively related to willingness to pay a higher price. In Kenya on the other hand, the perception of the SM brand among those aware of it was not a factor. This likely reason for this findings is that Kenyan free condoms users overall gave relatively low scores to the socially marketed brand (Mean = 3.43, SD = 3.52, out of 11 possible). These findings suggest that, as observed in previous studies, brand identification influences condom purchase decisions [11].

## 4. Discussion

As shown in recent studies [9], condoms remain an important component of the prevention of HIV, STIs, and unintended pregnancy. In 2017, UNAIDS estimated that 36.9 million people were living with the HIV virus, with 1.8 million newly infections, the majority through unprotected sexual intercourse. In addition, every day one million people are infected with curable STIs while over 500 million are already affected with HSV2 and 300 million with HPV [23]. At the International Conference on Family Planning in Nov 2018, the development community highlighted that 2014 million women and girls have an unmet need for contraceptives. Most these issues are more dramatic in SSA and among young and vulnerable people. Given the “youth bulge” in SSA that is projected to continue until the mid-21st century, there will eventually be more young people age 15–24 in the region than in India or China [4]. In part as a result of these demographic changes, estimates suggest that condom availability and use will represent the largest share of total HIV infections avoided by 2030 [24].

However, the condom funding and distribution landscape in SSA and elsewhere is changing rapidly. International and domestic funding for condom programming become has recently become stagnant or declined [1]. Another major change in the condom architecture is a reduction in donor support to social marketing organizations (SMO). Over the past 25 years, well-funded SMOs have developed effective condom distribution systems and promoted demand for condoms in SSA [25]. In 2014, USAID, the United Kingdom’s Department of Foreign Investment and Development (DfID), and other major donors began to phase out funding for social marketing programs, “graduating” brands and programs [26].

Given the changing donor and program landscape, this study is significant in several respects. First, based on nationally representative data, we found that free condom users are overwhelmingly willing to pay for condoms overall, and to an extent will pay prices that are higher than the prevailing rates in their country. This finding would support a potential expansion of social marketing efforts and commercial markets to reflect a larger pool of consumers who are able and willing to pay. In a landscape of declining donor funding for free condoms, this creates opportunities to use scarce funds for other important efforts such as campaigns to increase demand, branding of condoms, and coordination with commercial condom manufacturers to build a healthy total market approach for the product [27].

Second, we found that brand identification is associated with greater willingness to pay. This is consistent with the growing literature on branding of health and social change behaviors and products [28], and supports the idea that condom branding is an important demand creation strategy. The brand attributes sum measure used in our analysis provides some insight into the role of brand identification. These results are consistent with prior research on branding of socially marketed family planning products in low and middle income countries [25]. This study further highlights the competitive context of condom purchasing decisions where perceived brand equity of the competitor in comparison to the brand currently used could be an important consideration in the willingness to pay.

Finally, this study also reinforces the importance of free condoms. As expected, willingness to pay is inversely associated with wealth and thus ability to pay. Some countries, including South Africa and Zambia in this study, still have a majority of free condom users. While the makeup of the condom markets is changing, the reality is that free condoms are an important part of the overall HIV/AIDS prevention mix and this study reinforces the importance of maintaining free supplies. The challenge is to balance distribution and at the same time promote demand for paid condoms. Demand creation will be essential to create a healthy market where commercial condoms can become established in the context of decreased donor support for free distribution [29].

The current study has some limitations. First, while the data are nationally representative, they only reflect a single cross-section, so causality cannot be inferred and we are not able to model changes in willingness to pay and potential demand predictors over time. Second, there may be other important demand predictors, such as lifestyle factors and availability of other family planning and other products and services that influence condom uptake. Brand identification, in particular, was not the main focus of the study, and the survey had limited measures in this area. Previous research has found that measures of specific mental associations with benefits of condom brand use strongly predicts future behavior [14]. Third, while our findings demonstrate that there is substantial willingness to pay among free condom users, this does not automatically translate into purchase behavior, and future research is needed on strategies to increase condom purchasing.

Finally, we recommend that a greater proportion of future donor funding should go to demand creation programs, such as campaigns to raise awareness of the availability and affordability of paid condoms. This would increase the health of condom markets as sales and use across product categories would increase [11]. As new and expanded lines of condom products become available, campaigns should focus on promoting their adoption. Future research should follow consumers longitudinally to monitor effects of condom promotion efforts on use. Finally, experimental studies should test novel demand creation strategies, such as branding and use of digital technologies to reach consumers [30].

## 5. Conclusions

The landscape for condom markets in low and middle income countries is changing rapidly, with funding for free condom distribution declining. This research sheds light on factors including willingness to pay, branding, pricing, and distribution of condoms by country that will help future programs maximize the efficiency and effectiveness of condom marketing to fight HIV/AIDS, STIs, and serve a broad range of reproductive health needs. Future research should build upon the findings of this study and examine how best to promote condom use.

## Figures and Tables

**Figure 1 ijerph-16-00034-f001:**
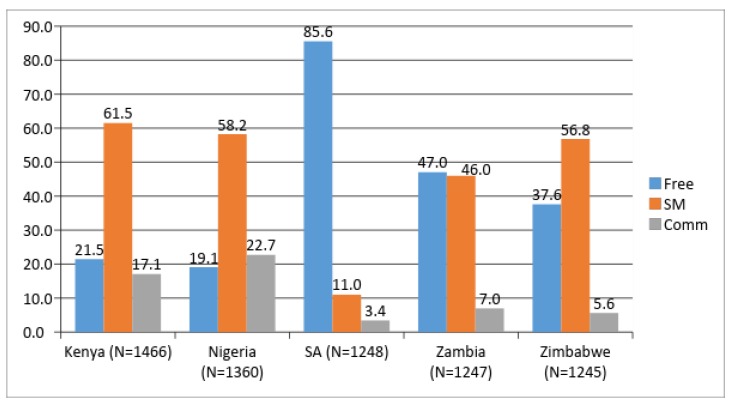
Summary of most recent condom use by country and source.

**Figure 2 ijerph-16-00034-f002:**
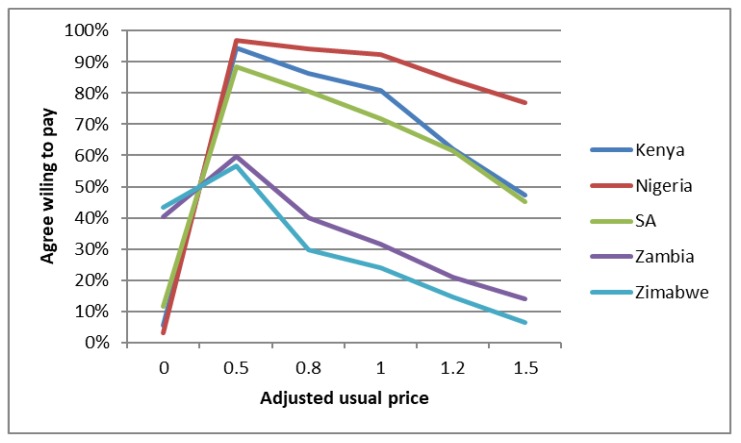
Free users’ price sensitivity (adjusted usual price is normalized across countries).

**Table 1 ijerph-16-00034-t001:** Will free condom users purchase condoms if free condoms are unavailable?

Usage Choice	Kenya (*N* = 589)	Nigeria (*N* = 488)	SA (*N* = 503)	Zambia (*N* = 496)	Zimbabwe (*N* = 500)
Stop using condoms	6.5	2.7	4.3	5.0	3.5
Go to another location where free condoms are usually available	16.2	23.0	63.2	25.7	40.8
Try to purchase condoms from stores nearby	69.9	69.9	30.0	62.4	38.4
Stop having sex	6.8	3.5	1.4	4.6	15.2
Other	0.6	1.0	1.1	2.2	2.1

**Table 2 ijerph-16-00034-t002:** Kenya willingness to pay (WtP) among free condom users: Aware and not aware of socially marketed (SM) brand.

Model 1: WtP All Free Users	Standardized Coefficients (beta)	*T*	Significance
Wealth (1 quintile difference)	−0.098	−8.051	0.000
Buy if no free	0.375	29.501	0.000
Convenience	0.092	7.489	0.000
Sum of free brand attributes	−0.036	−2.907	0.004
Dependent variable: Willingness to pay			
Model 2: WtP Free condom users aware of SM brand	Standardized coefficients (beta)	*T*	Significance
Wealth	−0.076	−5.623	0.000
Buy if no free	0.435	31.506	0.000
Convenience	0.065	4.757	0.000
Sum of free brand attributes	−0.077	−5.593	0.000
Dependent variable: Willingness to pay			

**Table 3 ijerph-16-00034-t003:** Zimbabwe willingness to pay (WtP) among free condom users: Aware and not aware of SM brand.

Model 1: WtP All Free Users	Standardized Coefficients (beta)	*T*	Significance
Wealth	−0.165	−13.962	0.000
Buy if no free	0.196	0.196	0.000
Convenience	−0.063	−5.529	0.000
Sum of free brand attributes	−0.305	−26.409	0.000
Dependent variable: Willingness to pay			
Model 2: WtP Free condom users aware of SM brand	Standardized coefficients (beta)	*T*	Significance
Wealth	−0.176	−12.833	0.000
Buy if no free	0.209	15.078	0.000
Convenience	−0.046	−3.534	0.000
Sum of free brand attributes	−0.265	−19.897	0.000
Dependent variable: Willingness to pay			
Model 3: WtP Free condom users aware of SM brand: Rank order of brand preference	Standardized coefficients (beta)	*T*	Significance
Wealth	−0.195	−13.371	0.000
Buy if no free	0.211	14.327	0.000
Convenience	−0.042	−2.993	0.003
Rank order of brand perception	0.188	13.175	0.000

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
