# Peer review of "Willingness to Pay for Condoms among Men in Sub-Saharan Africa"

_ijerph, 2018, doi:10.3390/ijerph16010034_

Round 1
Reviewer 1 Report
The research topic is relevant, it is well addressed and with interesting results. I suggest changing the title to "Willingness to pay for condoms in men from sub-Saharan Africa", since the study was performed only in men. One could understand that for a perhaps cultural issue it has been decided not to include women. I believe that this decision must be based in evidence and explained accordingly in the methods section. The promotion of self-care in STI´s prevention and pregnancy should not be discriminated by sex.
In the methods section, on the survey, it is not clear if it was asked about different prices in the local currency or with the increments of percentages explained, based on a standard or average price. The comment arises because, knowing the educational level of the African population, I find it difficult to understand the increases in percentages on the part of the respondents. It is important to explain this point better.
Finally, the second paragraph of the discussion (298-305) should be moved or merged into the Introduction section, since it repeats aspects already mentioned and is irrelevant in the discussion.
Author Response
We thank the reviewer for the valuable comments. See below specific responses:
Yes, we have changed the title of the paper as suggested.
We have clarified that the local currency was used to ask about willingness to pay in the survey instrument.
We have edited and moved portions of the paragraph at lines 298-305 as suggested.
Reviewer 2 Report
OVERVIEW
Thank you for this very interesting paper I enjoyed reading. It addresses an important topic – the willingness to pay for condoms in countries affected by HIV and the significance (or not) of cost and branding for individual choice. It’s clearly written and accessible.
I have only minor recommendations that I hope will make it suitable for publication.
INTRODUCTION
This provides a useful overview of the topic, and justification for the study. The section on social marketing is articulate and well presented, and it’s helpful to see the research questions front and centre.
One minor point to make (here, and throughout): ‘HIV/AIDS’ is generally not used. I’d recommend ‘HIV’ only. Also, there is a significant typo in the first sentence – STIs are of course ‘sexually’ transmitted infections, not ‘socially’ transmitted.
One thing I was not sure about – what’s a ‘SM brand’ of condom (line 80)?
METHODS
Sufficient detail is provided of the design and methods of data collection, and you explain in some depth the statistical tools that are used. I note that verbal consent was provided, and that ethical approval sought where possible.
The sampling process is well described, as is the data collection process. I couldn’t find a note in the paper as to whether or not you achieved your goal of 1,200 respondents in each country? I may have missed this, but it would be useful to be sure if it’s not mentioned.
RESULTS
This section is detailed, and provides useful insights into the findings. It’s certainly interesting that there are variations between countries in the source of condoms, use of free condoms, and willingness to pay. This is valuable information for policy makers dealing with donor transitions. It’s perhaps not surprising that the less wealthy are less willing to pay, therefore justifying the provision of at least someaccess to free condoms.
Was any disaggregation done on age? This would provide a useful additional insight.
DISCUSSION AND CONCLUSIONS
The discussion/conclusion is based suitably on the findings, and the summaries of the study’s main findings are succinct and specific, as are the recommendations. I note that you recommend further research, which is helpful, and acknowledge the study limitations, which are valid.
Review recommendations
- Major Compulsory Revisions
1. Tidy up ‘HIV/AIDS’ – make ‘HIV’
2. Correct typo on line 80
3. Clarify ‘SM brand’ - what is this?
4. Confirm final numbers of total respondents in each country
- Discretionary Revision
5. If data are available, it may be interesting to see if age was an influential factor.
Author Response
We thank the reviewer for the valuable comments. See specific responses below:
We corrected the misspelled 'sexually' in the introduction.
We clarified that SM refers to 'socially marketed'.
Regarding the sample size comment on methods, note that the Ns for each country are provided in the tables in the results section, and yes we achieved more than 1,200 sample size targets in each country.
Age was included in the multivariate models, and is thus accounted for in the analysis, as noted in the results section.
We thank the reviewer for several complimentary comments on the manuscript overall and again offer thanks for a positive review.